# Dimerization/Elimination of β-Styrylmalonates under Action of TiCl_4_

**DOI:** 10.3390/molecules28010270

**Published:** 2022-12-29

**Authors:** D. D. Borisov, G. R. Chermashentsev, K. V. Potapov, R. A. Novikov, Yu. V. Tomilov

**Affiliations:** N. D. Zelinsky Institute of Organic Chemistry, Russian Academy of Sciences, 47 Leninsky Prosp., 119991 Moscow, Russia

**Keywords:** styrylmalonates, (2-arylethylidene)malonates, dimerization/elimination, titanium tetrachloride

## Abstract

A new type of dimerization of dimethyl (β-styryl)malonates in the presence of TiCl_4_ accompanied by elimination of a methanol molecule was discovered. Selective methods for the synthesis of substituted trimethyl 4-hydroxy-[1,1′-biaryl]-3,3,5(2*H*)-tricarboxylates and trimethyl 7-hydroxy-9,10-dihydro-5,9-methanobenzo[8]annulene-6,8,8(5*H*)-tricarboxylates were developed. The regularities of the occurring processes were determined and a similar reaction of β-styrylmalonate with benzylidenemalonate in the presence of TiCl_4_ was performed in the scope of the suggested mechanism.

## 1. Introduction

β-Styrylmalonates **1** are isomers of 2-arylcyclopropane-1,1-dicarboxylates (ACDC, **2**) which, in turn, are the most common and accessible class of donor-acceptor cyclopropanes (DAC). The latter are widely used as versatile building blocks that make it possible to involve a three-membered ring along with donor and acceptor substituents [1,2,3,4,5,6,7,8]. To date, DACs proved to be useful synthons in complete syntheses of natural compounds. DACs can be used to obtain various functionally substituted compounds that have a wide range of chemical and biological types of activity [9,10,11,12,13,14,15,16,17,18,19,20,21]. In this context, the study of the chemical reactions of β-styrylmalonates can become a continuation of studies on ACDC chemistry and a relevant area of organic chemistry, since their behavior often differs from the reactions of ACDCs themselves. The efforts of our team allowed us to publish a series of works dealing with reactions of β-styrylmalonates **1** with aromatic aldehydes in the presence of various Lewis acids and under various reaction conditions. [22,23,24,25].

Similar to ACDCs themselves, β-styrylmalonates are highly reactive substrates. If no other substrates are present, ACDCs can undergo dimerization and oligomerization reactions [18,26,27,28,29,30,31,32,33,34,35,36]. As a rule, these processes are accompanied by partial isomerization of an ACDC into a styrylmalonate which, in turn, reacts with an activated ACDC molecule by generation of 1,3- or 1,2-zwitter-ionic intermediates [18,26,27,31,32]. It should be noted that dimerization is an interesting process in organic synthesis. As a rule, these reactions occur with high regio- and diastereoselectivity and in a single synthetic stage, which allows a range of both new and known compounds to be obtained. As of now, there are many works in literature that deal with ACDC dimerizations that can give both cyclization and annulation products (Figure 1) [18,26,27,28,29,30] and acyclic compounds [31,32,33]. In some cases, the cyclization process is accompanied by elimination of an alkoxy moiety, thus representing a dimerization/elimination reaction. Analysis of the structure of dimers shows that a styrylmalonate molecule is actually or formally involved in the formation of some of these dimers as active intermediates (there are even a few examples of cross-dimerization in reactions of ACDCs with styrylmalonate [28,34,35,36]). However, there is still no information on the possibility of any dimerization of β-styrylmalonates themselves. All starting β-styrylmalonates **1** were synthesized by a known method [37].

## 2. Results and Discussion

In this work, we discovered the first example of dimerization of β-styrylmalonates **1** in the presence of titanium tetrachloride accompanied by elimination of one alkoxy group. This process opens a way to substituted trimethyl 4-hydroxy-[1,1′-biphenyl]-3,3,5(2*H*)-tricarboxylates **2** and trimethyl 7-hydroxy-9,10-dihydro-5,9-methanobenzo[8]annulene-6,8,8(5*H*)-tricarboxylates **3** that are formed with involvement of one of the ester groups of the starting styrylmalonate. It is important to note that these dimerization/elimination processes can be partially controlled, as follows from the data on the optimization of these processes with styrylmalonate **1a** as an example (Table 1). It should be noted that in the presence of other Lewis acids we did not observe such type of the transformations [24]. Usually, we fixed oligomers or (4 + 2)-products in NMR spectra of the reaction mixtures. In fact, the reaction in the presence of 0.5 equiv. TiCl_4_ in 1,2-dichloroethane under reflux conditions gives almost exclusively cyclohexadienol **2a**, whereas the reaction with excess TiCl_4_ (1.5 equiv.) at a lower temperature (in dichloromethane, reflux conditions) gives substituted dihydro-5,9-methanobenzo[8]annulene **3a**, though its maximum yield is as small as 30%. The relatively low yield of compound **3a** is explained by competitive side processes, in particular, (4 + 2)-dimerization and oligomerization.

In order to study the effect of electronic and steric factors on the reactions, the dimerization reaction was performed with a number of substituted β-styrylmalonates. It was found that β-styrylmalonates with an acceptor substituent in the aromatic ring as well as various halo-substituted derivatives readily enter the process under study on heating in dichloroethane, while the position of the halogen atoms in the aromatic moiety does not significantly affect the occurring reactions. The process with these substrates occurs quite selectively without significant formation of side products. Moreover, the final compounds, i.e., the corresponding 2-hydroxycyclohexa-2,4-diene-1,1,3-tricarboxylates **2a–e**, do not require additional purification after extraction. The sterically hindered β-styrylmalonate and a styrylmalonate with a donor substituent in the aromatic ring underwent this type of dimerization somewhat less readily (Figure 2). The ^1^H NMR spectra of the reaction mixtures contained a significant number of dimers **4f,g** formed by (4 + 2)-annulation, as we reported previously [18]. Moreover, the donor moiety in styrylmalonate **1f** partially favors yet another reaction pathway to give not only compound **2f** (Figure 2) but also a small amount of substituted dihydro-5,9-methanobenzo[8]annulene **3f** (Figure 3).

Subsequently, we studied a deeper transformation of substituted β-styrylmalonates that resulted in dihydro-5,9-methanobenzo[8]annulenes **3**. It was found that a narrower range of substrates could be used in this process compared to the formation of cyclohexadienols **2** due to the high sensitivity of the reaction to the position and nature of the substituent in the aromatic part of styrylmalonate. The reaction occurs rather successfully with styrylmalonate **1a** itself or with its *meta*-bromo substituted derivative (Figure 3). Although the presence of a substituent, e.g., a fluorine atom, at the *para*-position of the aromatic moiety makes it possible to obtain a certain amount of dihydro-5,9-methanobenzo[8]annulene **3e**, the typical formation of the dimeric product of (4 + 2)-annulation **4e** is still the main process. A donor substituent in the aromatic moiety, similar to sterically hindered naphthyl, shifts the reaction pathway of these styrylmalonates toward the formation of (4 + 2)-annulation products **4f,g**. Nevertheless, the annulated compound **3f** was obtained in small yields by the reaction of styrylmalonate **1f** with 0.5 equiv. TiCl_4_ under the conditions used to synthesize cyclohexadienol **2f** (Figure 3). An acceptor substituent at the *para* position does not favor the formation of fused rings **3**, either. For example, the action of TiCl_4_ (1.5 equiv.) on (4-nitrostyryl)malonate **1c** results in intense reddish-brown coloring of the reaction mixture with formation of a significant amount of oligomers among which it was almost impossible to identify any dimerization products. In contrast, though the formation of dihydro-5,9-methanobenzo[8]annulene **3d** was not observed in the case of (2-chlorostyryl)malonate **1d**, almost no oligomerization processes were observed either, and according to the NMR spectra of the reaction mixture, all the major signals corresponded only to compound **2d**.

9,10-Dihydro-5,9-methanobenzo[8]annulenes **3** are formed as a single diastereomer where the aryl substituent is oriented toward the OH group (Figure 1). At the same time, the other diastereomer is not detected in any noticeable amounts.

In addition to the dimerization/elimination of β-styrylmalonates **1** in the presence of titanium tetrachloride, a cross variant of a similar process was carried out by the reaction of β-styrylmalonate **1a** with benzylidenemalonate **5** as one of the possible components for the formation of cyclohexadienols **2**. In fact, a mixture of two different cyclohexadienols in 1: 2 ratio was obtained in the reaction of malonates **1a** and **5** in 1: 2 ratio in the presence of 0.5 equiv. TiCl_4_ at 80 °C. Cyclohexadienol **2a**, a product of formal dimerization/elimination of styrylmalonate **1a** described above, was a minor compound, while the related cyclohexadienol **6a** formed by the reaction of benzylidenemalonate **5** with styrylmalonate **1a** and also with elimination of a methanol molecule was the main compound according to NMR and mass spectra (Figure 4). It should be noted that due to similarity of the structures of compounds **2a** and **6a**, we failed to separate them completely and to isolate cyclohexadienol **6a** individually even after double chromatography on SiO_2_. An attempt was made to perform the reaction with a large excess of benzylidenemalonate **5**, but even in this case we failed to avoid the formation of homodimer **2a** completely.

The same reaction of benzylidenemalonate **5** with (4-methylstyryl)malonate **1f** occurred more selectively. According to the ^1^H NMR spectra of the reaction mixture, the homo-dimerization/elimination product **2f** was detected in trace amounts, whereas the cross-coupling product **6f** (**2f**/**6f** ratio approximately 1:16) predominated (Figure 4). The total yield of cyclohexadienols **2f** and **6f** was smaller than in the case of unsubstituted styrylmalonate **1a** and was approximately the same as in the homo-dimerization/elimination reaction of (4-methylstyryl)malonate **1f** (see Figure 2), which is still due to the fact that its dimerization to give the (4 + 2)-annulation product **4f** occurs more readily.

Finally, we tested the variant of the asymmetric reaction of dimerization/elimination of β-styrylmalonates, for which no asymmetric processes have been described in the literature so far. In fact, this turned out to be not an easy task, since the standard asymmetric catalytic approaches using chiral ligands, which were used for ACDC reactions [38,39,40,41], would hardly work for the processes under consideration under conditions of equimolar amounts of strong Lewis acids. As a result, after a series of experiments, we focused not on the elusive development of a catalytic variant with chiral ligands, but on the use of chiral substituents in ester groups. As such a fragment, we used an available natural (–)-menthyl substituent [42] and synthesized the corresponding styrylmalonate **1h** (according to the standard method from ACDC). Under the influence of 1.5 equiv. TiCl_4_ under standard conditions, the menthyl styrylmalonate **1h** also enters the dimerization/elimination reaction with the formation of dihydromethanobenzoannulene **3h** (Figure 5), while the cyclohexadienol derivative **2h** is not formed even when using 0.5 equiv. TiCl_4_. The steric effect of the substituents in the ester groups undoubtedly affects the efficiency of the process, and the yield of tricycle **3h** turns out to be low, only 18%, but “e.r.” turns out to be good enough (9:1) for such a simple chirality induction, which demonstrates the fundamental efficiency of this approach.

Taking these results into consideration, the following mechanism of formation of cyclohexadienols **2** may be assumed. At the first stage, β-styrylmalonate **1** is activated with titanium tetrachloride and two different intermediates **I** and **II** are generated (Figure 6), which seems to be especially favored by the use of 0.5 equivalents of TiCl_4_. Subsequently, ionic (4 + 2)-cycloaddition of these intermediates occurs to give a polyfunctionally substituted cyclohexene **III**, which is converted to titanium enolate **IV** due to elimination of a methanol molecule. The ability to react with Knoevenagel adducts of type **5** (intermediate **I**) is an additional evidence that the reaction occurs as formal (4 + 2)-cycloaddition (Figure 7). At the final stage of the process, hydrolysis results in the final cyclohexadienols **2**. It should be noted that we did not observe the aromatization of compounds **2**. At the same time, in the case of self-condensation of other 1,3-dicarbonyl compounds, in particular enaminodiones or diethyl 2-ethoxymethylenemalonates, the formation of substituted benzene compounds is observed [43,44].

The mechanism of formation of benzobicyclo[3.3.1]octenes **3**, which, strangely enough, are formed at a lower temperature but with an excess of TiCl_4_, appears to be more complex. Control experiments with compounds **2a** and **4a** showed that the presence of 1.5 equivalents of titanium(IV) tetrachloride in boiling dichloromethane did not result in the formation of any amounts of compound **3a**. The NMR spectra of the reaction mixtures after acid treatment contained only signals of the initial compounds, and thus we can assume that the pathway leading to compound **3a** is established already at the first stages of the process. In this case, titanium tetrachloride is apparently coordinated to the malonyl moieties of both intermediates **I** and **II**. As a result, annulation to the aromatic ring of **V** occurs first, followed by cyclization with involvement of malonyl moieties to form bicyclic titanium enolate **VI** (Figure 8).

## 3. Experimental Section

See Appendix A.

## 4. Conclusions

Thus, we have discovered a new previously unknown variant of dimerization of β-styrylmalonates **1** that is accompanied by elimination of an alcohol molecule and is not observed in the case of isomeric ACDCs. Titanium tetrachloride acts as a catalyst. The process we discovered is quite general for various substituted β-styrylmalonates and, as a rule, gives high yields of the corresponding 2-hydroxycyclohexa-2,4-diene-1,1,3-tricarboxylates **2**, up to 93%. A variant of controlling the dimerization process to reach deeper transformations by varying the amounts of TiCl_4_ and reaction temperature that gave 9,10-dihydro-5,9-methanobenzo[8]annulenes **3** was suggested. The effect of electronic and steric factors on the observed process was noted: halo-substituted styrylmalonates and styrylmalonates containing an acceptor moiety in the aromatic nucleus successfully undergo this reaction. Moreover, the principal possibility to perform similar reactions with Knoevenagel adducts **5** was shown.

## Data Availability

The data presented in this study are available on request from the corresponding author.

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
