# Peer review of "Dimerization/Elimination of β-Styrylmalonates under Action of TiCl4"

_molecules, 2022, doi:10.3390/molecules28010270_

Round 1

Reviewer 1 Report

The work of Tomilov et al. represents an interesting approach to the preparation of polyfunctional cyclohexene derivatives based on styrylmalonates. The article is well and thoroughly described with the use of spectral data and raises no doubts about the validity of the presented structures. Overall, this manuscript will be of interest to researchers in the field of the polysubstituted carbocycles synthesis. If we take into account the possibility of further transformations of the compounds obtained by Tomilov et al. the way of aromatization of the derivatives is obvious.  Several questions and wishes arose while reading this manuscript. Firstly, reading the work there were parallels in the synthesis of arenes based on trimerization or dimerization of various derivatives of 1,3-dicarbonyl compounds (DOI: 10.1016/j.tetlet.2012.02.015; DOI: 10.1021/acs.joc.9b00623). It would be good to mention these works to show the peculiarities of cyclizations from the chosen derivatives of 1,3-dicarbonyl compounds. Secondly, the choice of titanium(IV) chloride is not discussed in the article. Have there been attempts to use other catalysts based on silicon, boron? If such preliminary experiments were conducted, it would be worth mentioning in the text. I wonder if the selectivity of the presented transformations such as in Scheme 4 can be controlled by means of extra addition of the base? Have attempts been made to obtain aromatic derivatives based on compounds of type 2?

Author Response

Dear Reviewer,

Thanks for your comprehensive review and positive evaluation. Your remarks were useful for us and for improving of the quality of manuscript. We have considered your remarks and recommendations and have tried to correct them on all points. The detailed list and discussions are given below:

  1. We added the data of di- and trimerization of 1,3-dicarbonyl compounds in the mechanistic part of our work. We compared the aromatization of these compounds with our substrates.
  2. We studied the reactivity of β-styrylmalonates in the presence of various Lewis acids in our previous work (DOI: 10.1021/acs.joc.0c02891). In case of using other catalysts based on silicon, boron, we observed the formation only [4+2]-products such as product 4 in the manuscript. The addition of the base in the reaction mixture was stopped the reaction, we observed only starting materials in spectra of the reaction mixtures. We made an attempt to obtain benzene derivatives from compounds 2 using DDQ, but the reaction failed. We observed only the signals of compound 2 in the spectrum of the reaction mixture.

Best regards, Dr. Roman Novikov

Reviewer 2 Report

In this manuscript, Timilov and coworkers reported A new dimerization reaction of dimethyl (β-styryl)malonates promoted by TiCl4  for the synthesis of trimethyl 4-hydroxy-[1,1´-biaryl]-3,3,5(2H)-tricarboxylates and trimethyl 7-hydroxy-9,10-dihydro-5,9-methanobenzo[8]annulene-6,8,8(5H)-tricarboxylates. The reaction mechanism was also proposed. The current TiCl4-promoted annulation method should be interesting to the practitioners of organic synthesis. However, several important issues need to be addressed before the manuscript can be accepted for publication.

1. In Scheme 1, references should be labeled for each type of known cyclodimerizations.

2. In Table 1.Entry 9 and 10, what is the difference between the two?

3. In Scheme 7 and Scheme 8, the mechanisms were described too roughly to understand. The formation and break of each bond should be drawn in good orders.

4. In Scheme 7 and Scheme 8, TiCl3 does not have to be hydrolyzed to be removed from the enol. The complex is not that stable. Ligand exchange just can do the work.

5. Be honest, the NNR spectra of all products are not of high quality. A significant amount of impurities could be seen. It is the authors' job to guarantee the purity of the samples. Size exclusion gel chromatography (Sephadex LH-20) could be a possible solution to remove the impurities. To prove the existence of the claimed products, the authors should provide the original HRMS spectra of all compounds characterized.

Author Response

Dear Reviewer,

Thanks for your comprehensive review and positive evaluation. Your remarks were useful for us and for improving of the quality of manuscript. We have considered your remarks and recommendations and have tried to correct them on all points. The detailed list and discussions are given below:

1. We added the references in the scheme as you wish.

2. The difference between entry 9 and 10 in Table 1 in the reaction time.

3. and 4. We make changes in Scheme 7,8 as you wish.

5. We added the HRMS data in the Supplementary as you wish.

Best regards, Dr. Roman Novikov